# Characteristics of ZrC Barrier Coating on SiC-Coated Carbon/Carbon Composite Developed by Thermal Spray Process

**DOI:** 10.3390/ma12050747

**Published:** 2019-03-05

**Authors:** Bo Ram Kang, Ho Seok Kim, Phil Yong Oh, Jung Min Lee, Hyung Ik Lee, Seong Min Hong

**Affiliations:** 1High-enthalpy Plasma Research Center, Chonbuk National University, 546 Bongdong-ro, Bongdong-eup, Wanju-gun, Jeonbuk 55317, Korea; brkang@jbnu.ac.kr (B.R.K.); hoseok@jbnu.ac.kr (H.S.K.); philoh@jbnu.ac.kr (P.Y.O.); 2Agency for Defense Development, Yuseong PO Box 35, Daejeon 305600, Korea; junglee@add.re.kr (J.M.L.); hyungic@add.re.kr (H.I.L.)

**Keywords:** carbon/carbon (C/C) composites, ultra-high temperature ceramic (UHTC), vacuum plasma spray (VPS), ablation resistance

## Abstract

A thick ZrC layer was successfully coated on top of a SiC buffer layer on carbon/carbon (C/C) composites by vacuum plasma spray (VPS) technology to improve the ablation resistance of the C/C composites. An optimal ZrC coating condition was determined by controlling the discharge current. The ZrC layers were more than 70 µm thick and were rapidly coated under all spraying conditions. The ablation resistance and the oxidation resistance of the coated layer were evaluated in supersonic flames at a temperature exceeding 2000 °C. The mass and linear ablation rate of the ZrC-coated C/C composites increased by 2.7% and 0.4%, respectively. During flame exposure, no recession was observed in the C/C composite. It was demonstrated that the ZrC coating layer can fully protect the C/C composites from oxidation and ablation.

## 1. Introduction

Carbon/carbon (C/C) composites have low weight, high thermal shock resistance, and good ablation resistance at high temperatures, which facilitate their use in engines, nose tips, leading edges, nozzles, and thermal protection systems for space vehicles operating in severe environments, i.e., during hypersonic flight, atmospheric re-entry, and propulsion [1,2]. However, the working conditions of the abovementioned systems are severe. For example, re-entry vehicles can experience high temperatures of over 2000 °C, which is sustained for several tens to hundreds of seconds with high-pressure air and high-velocity particle erosion. However, C/C composites are vulnerable to oxidation below 400 °C and undergo ablation during high-speed gas jet and particle erosion, which restricts their application in this field. Hence, much effort has been undertaken to improve the ablation resistance of C/C composites for application at higher temperatures and in oxygen-containing atmospheres. 

In recent years, extensive research was conducted with the aim of applying ultra-high temperature ceramics (UHTCs) to C/C composites, owing to their high melting point (>3000 °C) and good thermal properties [3,4,5,6]. Coating methods are widely used to protect C/C composites owing to the simplicity of the fabrication and integrity with the substrate. UHTCs including transition metal carbides and borides can improve the ablation resistance and ablation properties of C/C composites by doping into the carbon matrix or coating on the composite surface [7,8,9,10,11]. Among UHTCs, ZrC is one of the most promising candidates due to its high melting point of over 3400 °C [12,13]. Additionally, its oxidation product (ZrO_2_) has a relatively high melting point of 2700 °C and low vapor pressure; hence, it can form a protective layer on the surface of the C/C composites and reduce the oxygen diffusion rate.

The conventional method for ZrC coating is based on chemical vapor deposition (CVD) [14,15], however, it is complex, expensive, time-consuming, and hazardous to the environment. An alternative low-cost method is chemical vapor infiltration (CVI) [16], however, it has the disadvantage of low efficiency. Both are well developed and widely used to cover the C/C composites by dense ZrC layers, but limit the shape and size of the substrate. Therefore, it is necessary to identify a suitable approach to coat on the C/C composites. 

Plasma spraying is one of the most promising coating methods because it is economical and can be easily applied on an industrial scale [17,18,19]. It is also a suitable technique for depositing carbides with high melting points such as ZrC, HfC, etc. [20,21]. In the VPS (vacuum plasma spray) process or in other plasma spray processes in the atmosphere, ZrC powder is oxidized to result in the formation of zirconium oxides [22,23,24,25]. Therefore, VPS coatings are especially preferred because of their high purity with low porosity and high deposition rate without the formation of oxides [26]. In VPS, the coating layer is formed by melting the constituent particles at a very high temperature and by colliding them with the base material. Therefore, the inherent properties of both components, i.e., the molten particles and base substrate, are preserved. However, as numerous parameters contribute to the formation of a high-quality coating layer, it is important to determine the optimal process parameters. Among the operational parameters, it is well known that the velocity and temperature of the plasma jet, which can be controlled mainly by the discharge current, are closely associated with the phase change, thickness, and porosity. As mentioned above, although the coating process using VPS has many advantages, few research articles have reported on UHTCs coatings prepared using the VPS technique. 

Therefore, in this work, we investigated the discharge current effects for a ZrC coating layer on a C/C composite and evaluated the thermal ablation performance of the coating. Scanning electron microscopy (SEM) and X-ray diffraction (XRD) analyses were performed to analyze the coating layer properties. Furthermore, an ablation test was conducted to determine the ablation resistance of coating layers fabricated under various current conditions of VPS. 

## 2. Materials and Methods 

### 2.1. ZrC Coating Process on C/C Composites

Disc-shaped C/C composites (Dai-Yang Industry Co.) with a 5 mm thickness and a 30 mm diameter were used as the substrates. To reduce the difference in the thermal expansion coefficient between ZrC and the carbon composites, a thick SiC layer (approximately 30 µm) was fabricated on the top surface of the C/C composite by chemical vapor reaction (CVR) [27]. ZrC powders (D50 = 8 µm; purity >99.9%, Avention, Korea) were used to coat the ZrC layer using a VPS system (Oerlikon Metco AG, Wohlen, Switzerland) with a F4-VB gun (Oerlikon Metco AG, Wohlen, Switzerland). The ZrC raw powders exhibited irregular shapes (Figure 1). 

Prior to the coating process, the substrates were installed in the chamber and aligned perpendicular to the gun. Figure 2 shows the three main steps involved in the coating process. The first step is pre-heating the substrates, which increases the liquidity of the substrate surface and improves the adhesion flexibility of the surface with the molten ZrC droplets in the next coating step. Moreover, impurities on the surface are removed in this step. The splat shape of the powder varies depending on the pre-heating temperature. As the temperature increases, the radius of the splats increases, however the droplets spread too wide, thereby hindering the formation of the coating. Therefore, it is important to choose the appropriate pre-heating temperature. The distance between the substrate and plasma gun was fixed at 210 mm and the pre-heating was repeated along an S-shaped path. The movement of the plasma gun from the top left to the bottom right of the substrate and then back to the top left was defined as one cycle, with a total of five cycles performed. Throughout the pre-heating process, the surface temperature of the substrate was maintained below 900 °C. 

In the second step, the coating process (Figure 2b), the ZrC powder from the injector was melted by the plasma flame, and the droplets were attached on the substrates with a high speed. In this step, the distance between the substrates and plasma gun was adjusted to 350 mm. Ar and H_2_ were used as the plasma forming gases, and Ar was also used as the powder carrier gas. The powder feeding rate was 4.5 g/min. To obtain a coating layer with uniform thickness, coating was performed over an S-shaped path, similar to the pre-heating process. Coating was performed at 10 mm intervals and in all, 20 coatings were applied. The current was varied as 500 A, 550 A, 600 A, 650 A, and 700 A in the experiment, and the coated sample was denoted as ZS-1, ZS-2, ZS-3, ZS-4, and ZS-5, respectively. Table 1 shows the detailed process conditions. 

Finally, the specimens’ temperatures sharply decreased after the coating process. At this moment, post-heating was applied to the specimens to prevent crack/defect formation, and to prevent exfoliation of the coating layer due to the difference in the thermal expansion coefficient between the substrates and the ZrC coating. Post-heating treatment was performed under a lower current (500 A) compared to the current used in the pre-heating and coating steps, so as to prevent damage to the coated specimens. The total time taken for coating was around 5 min. 

### 2.2. Coating Characterization

After cutting the five coated samples, their surfaces were polished using 1 μm of diamond paste to analyze the characteristics of the coating layer. Phase analysis of the coating was conducted by XRD (D8 Advance, Bruker, USA) at 40 kV, with Cu-Kα radiation, and a scanning speed of 4 °/min. The cross-sectional microstructure of the coating was determined by field emission scanning electron microscopy (FE-SEM; SU-8030, Hitachi, Japan), and the components of the coating layer were analyzed by energy dispersive X-ray spectroscopy (EDS; X-MaxN80, Horiba, Japan). Table 2 shows the results of the analysis for the coating layer. The thickness and porosity were determined using an image analyzer by referring to the FE-SEM images and by porosity measurements based on ASTM E-2109 (Test Methods for Determining Area Percentage Porosity in Thermal Sprayed Coatings). Additionally, the coating layer’s bonding strength and porosity were analyzed. Finally, the ablation test was carried out for the ZrC-coated sample, which was selected based on the best result in metallurgical analysis. Before the ablation test, an adhesion test of the selected sample was also carried out by the universal testing machine (UTM; 5982, Instron, USA). Figure 3 shows a schematic of the adhesion test with the sample fixed with glue (Fusionbond 370, Hernon, USA) between the bars.

### 2.3. Ablation Test with Supersonic Flame

The ZrC coating layer was expected to protect the C/C composites in a high-temperature oxidizing environment. Therefore, the weights and thicknesses of the sample before and after ablation were measured by an ablation test, and the ablation rate was calculated. The ablation characteristics of the coated and uncoated samples were compared, and the performance of the coating layer was verified based on the results. 

Among the five coated samples (ZS-1~ZS-5), the sample with the best characteristics was selected for the ablation test. A high-velocity oxygen fuel (HVOF) system equipped with a diamond jet (DJ) gun (DJH2700 gun; Oerlikon Metco, United States of America) was used. The DJ gun uses a combination of oxygen, fuel gas, and air to generate a high-pressure flame with uniform temperature distribution. A pre-mixed fuel gas (typically, propane, methane, propylene, or oxygen) and oxygen come into contact with the supplied air to generate a high-temperature combustion gas. The temperature of the generated flame approaches 2,730 °C, and the flame is accelerated through convergent/divergent nozzles to create a supersonic flame. In this work, commercial liquefied petroleum gas (LPG) was used as the fuel. The pressure and flow velocity were 10.3 bar and 22 L/min for oxygen, 6.2 bar and 7.5 L/min for the fuel, and 7.2 bar and 58 L/min for air. As shown in Figure 4, the specimen was installed 60 mm away from the nozzle and was aligned perpendicular to the nozzle. The ablation test was conducted for 30 s at temperatures higher than 2,230 °C in open air. During ablation, the surface temperature of the specimen was measured using a two-color pyrometer. We used a digimatic micrometer (Mitutoyo), a non-contact 3D surface measuring system (IFM G4, Alicona), and a 0.1 mg-precise electronic balance for before and after test evaluation of the recession and mass loss of the test samples. The linear and mass ablation rates of the sample were calculated using the following equations:(1)$$R_{m}=\frac{\Delta m}{t}$$
(2)$$R_{l}=\frac{\Delta l}{t}$$
where, *R_m_* is the mass ablation rate, *Δm* is the mass change before and after ablation, *R_l_* is the linear ablation rate, *Δl* is the change of the thickness at ablation region, and *t* is the ablation time. 

## 3. Results and Discussion

### 3.1. Microstructure and Phase of the As-Sprayed Coatings

Figure 5 shows the samples’ FE-SEM cross-sections before and after the ZrC coating layers by different conditions. Figure 5a shows that SiC is well bonded on the C/C composite substrate, with a 30 µm thickness. The coated samples (Figure 5b–f) show different coating layer characteristics depending on the discharge currents. ZrC is uniformly coated on top of SiC without any loss of SiC, however some cracks formed by in the coating layer of samples during the cutting. The thickness of the coating layers gradually increases from 75 to 122 µm as the discharge current increases. While the thickness increases, more cracks were observed. As cracks formed during the cutting for cross-sectional observation due to the accumulated thermal stress, a partial delamination from the substrate was observed in a thicker case, as shown in Figure 5f.

Figure 6 shows high-magnification FE-SEM images, detailing the interface between the ZrC and SiC layers. The light gray region represents the ZrC layer, while the dark gray region represents the SiC layer. There is no mixed region between the ZrC layer and the C/C substrate, hence, each layer is distinct. Figure 7 shows the results of the porosity measurement. Although there is no significant change in the porosity, the porosity slightly increases with the current with the exception of ZS-2. The discharge current could significantly affect the velocities and temperatures of the particles as well as the thermal/physical characteristics of the plasma flow. If the current increases, the discharge power, which can bring about a higher gas temperature of the plasma jet, also increases, therefore, more powders can be melted. Splashing occurs due to poor spreading of the particles on the substrate surface. Therefore, the porosity increases as the molten particles splash in the substrate under a high current. For the ablation test, the above results were considered and the ZS-2 sample was selected as the one with the best characteristics. Figure 8 shows the results of the phase analysis of the coated sample. Figure 8a shows the EDS mapping results of the ZS-2 sample. In the upper part of the figure, light blue represents Zr, green represents Si, and red represents C. Figure 8b shows the XRD diffraction pattern of the coated sample fabricated using VPS. As shown in Figure 8b, it is clear that there are strong peaks corresponding to ZrC, while diffraction peaks of carbon are not detected, indicating that the thick ZrC layer is formed on the C/C composite. Meanwhile, narrow and sharp peaks indicate good crystallization of the ZrC phase. The peaks at 2θ = 33°, 36°, and 56° match well with those of ZrC corresponding to lattice planes of (111), (200), and (220) according to the Joint Committee on Powder Diffraction Standards (JCPDS) database [11-0110]. From the result of the XRD and EDS analyses, SiC and ZrC are well identified, and confirm that ZrC is well coated on the substrate without the formation of oxides or the influx of impurities.

To perform the ablation test, we selected the ZS-2 condition because of its obtained low porosity and uniform thickness without cracks and delamination. Using this condition, we increased the number of coating cycles to obtain thicker layers of up to 163 μm, which should be useful for prolonged ablation. Figure 9 shows the results of the adhesion test on the ZS-2 sample. Figure 9a shows tensile strength and Figure 9b shows the tested sample ZS-2. As a result, the ZrC layer was not separated with the SiC layer, however it was separated with the P1. Furthermore, the ZrC layer was not peeled from the substrate. Based on these results, it can be seen that ZrC is well bonded with SiC.

### 3.2. Ablation Properties

Table 2 shows the ablation properties of the uncoated and ZrC-coated C/C composites. Note that there is an obvious weight loss in the ablation process for the C/C composite, while the coated sample gains weight. For the specimen without the coating, the weight decreases by 0.95 g. The thickness of the specimen also decreases however, it was difficult to accurately measure the ablation rate as the diameter of the flame was smaller than that of the specimen and consequently, ablation was not uniform. Therefore, the ablation rate of the specimen was calculated using a non-contact 3D surface measuring system (IFM G4, Alicona). Figure 10a shows the surface image of the ablated specimen, from which the ablated length was calculated using the difference in heights before and after the ablation test, which is shown in Figure 10b. The most ablated part was measured and determined as the linear ablation rate. On the other hand, for the coated specimen, the weight and thickness increased by 0.8 g and 113 µm, respectively. Figure 11 shows the results of XRD analysis of the specimen surface after the ablation test and ZrO_2_ component. Only the oxidation of ZrC is observed. It appears that the outer ZrC layer can be transformed into ZrO_2_, which results in weight and volume increases due to the formation of the oxide. The ablation test was performed for 30 s, and the maximum surface temperatures of the coated and uncoated specimens were measured to be 2,052 °C and 2,275 °C, respectively.

### 3.3. Morphology Analysis of the Coatings After Ablation

Figure 12 shows images of the uncoated and coated C/C composites before and after ablation. In the uncoated C/C composite, the center is subjected to the maximum ablation. Furthermore, the surface color of the coated specimen changes from black to white by oxidation.

Figure 13a,c show the surface morphology of the uncoated C/C composite before ablation, while Figure 13b,d show the surface morphology after ablation. After ablation, several large pores generated on the surface, which show irregular patterns as seen in Figure 13b. Additionally, the spaces between the fibers are filled with carbon matrices (Figure 13c), however after ablation, the carbon matrices disappear and the carbon fibers are exposed with a decreased thickness (Figure 13d). Furthermore, the fibers show a tapered shape that indicates that the carbon matrices are first oxidized, followed by the fibers, due to which the tips of the fibers are deformed into a needle shape (Figure 13d). 

Figure 14 shows images of the specimen’s surface before and after the ablation test. Figure 14a shows the top of the ZrC coating layer before ablation, wherein cracks and pores are not observed. Figure 14b shows the center area in the ablation specimen, from which it can be seen that the ZrO_2_ layer has completely melted. This is confirmed in Figure 12d, wherein the cracks seem fine on the entire surface, from the center of the specimen after ablation, and microcracks and micropores are observed from the surface, as shown in Figure 14b. It appears that such cracks and defects occurred due to the difference in the thermal expansion coefficient caused by a sudden thermal shock. Furthermore, it seems that many microcracks and micropores generated on the specimen surface because the ZrO_2_ formed on the specimen surface underwent a phase change into tetragonal zirconia at a high temperature, which further changes into monoclinic zirconia upon cooling after completion of the ablation test. Therefore, as mentioned, cracks seem to have formed due to the volume change and rapid cooling caused by the two phase changes, and pores seem to have formed due to CO or CO_2_ gas, which was generated when ZrC reacted with oxygen.

Figure 15 shows FE-SEM images and EDS mapping analysis of the specimen’s cross-section after the ablation test. Figure 15a shows the cross-section of the coated specimen after the ablation test. Part of the coating layer appears to be exfoliated when the specimen was cut for observation. However, the bottom of Figure 15a shows that the remaining ZrC layer is still well bonded on the SiC layer. At the top of the coating layer, a uniform oxide layer with over 30 μm thickness formed. From XRD analysis (Figure 11), the coating layer is covered by ZrO_2_. Additionally, the ZrC layer and substrate are well bonded at the interface without exfoliation after the ablation test. Figure 15b,c show the image and results of the EDS line scan profile and point analysis, respectively. The upper and middle parts of the coating layer, and the substrate beneath the coating layer were selected for analysis. The white phase in spectrum 1 is primarily ZrO_2_ produced by ZrC oxidation, and the light gray phase in spectrum 2 and spectrum 3 is mainly ZrC and ZrO_2_. The gray phase in spectrum 4 is mainly SiC with no SiO_2_ detected. The proportion of oxygen decreases toward the substrate. SiO_2_ is not observed as the oxygen could not spread to the SiC layer that forms an interface with ZrC. Figure 15c shows the results of the sample’s line scan analysis. Carbon peaks are observed, because the resin flows through the gap in the middle of the coating layer. Oxygen peaks are observed with a high intensity at 70–80 µm depth, which weakens thereafter. Based on the EDS and previous XRD results, it is clear that the ZrO_2_ protective film on the surface of the coating layer effectively prevents oxygen diffusion from ZrC to the C/C composite. Furthermore, the ZrO_2_ layer serves as a barrier to retard thermal diffusion and thus, reduced heat transfer to the underlying C/C composites.

## 4. Conclusions

We developed an optimal ZrC coating process by changing the discharge current in a VPS system for improving thermal stability of a C/C composite. The coating was performed at different current values that ranged from 500 to 700 A, and the ablation test was conducted for the best condition of the ZrC coating using a HVOF system. Thick ZrC layers were coated on the C/C composite, where SiC layers formed by CVR served as the interlayer between them. By increasing the discharge current, the coated thickness and porosity also increased. At a higher current condition, cracks and delamination of the ZrC layer commenced. During the ablation test, the C/C composite was completely protected from thermal oxidation by the thick ZrC layer which formed ZrO_2_. After the ablation test, although cracks and pores were observed from the surface of the coating layer, no delamination was observed at the coating layer interface. Phase analysis by XRD and EDS indicated that the ZrO_2_ formed at the top of the coating layer was transferred into the C/C composites. Thus, the coating process using VPS effectively protected a C/C composite in an ablation environment; moreover, a detached coating was not detected between the ZrC coating and substrate, indicating good adherence between them.

## Figures and Tables

**Figure 1 materials-12-00747-f001:**
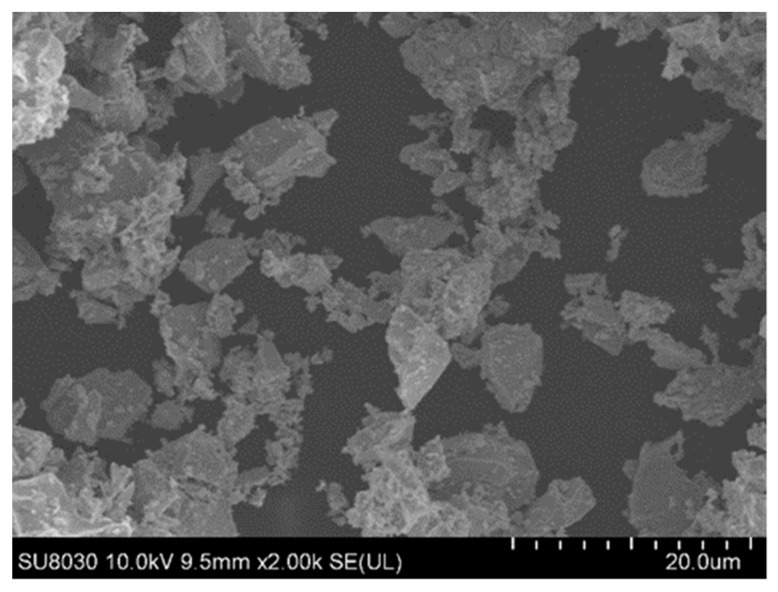
Field emission scanning electron microscopy (FE-SEM) morphology of ZrC powder.

**Figure 2 materials-12-00747-f002:**
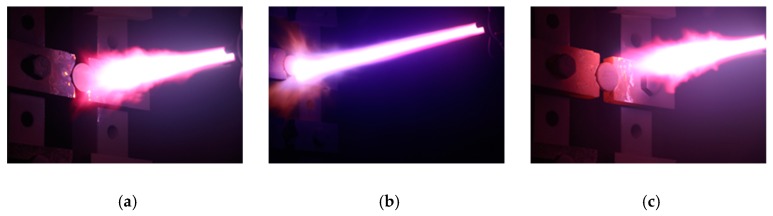
The coating process: (**a**) Pre-heating, (**b**) coating, and (**c**) post-heating.

**Figure 3 materials-12-00747-f003:**
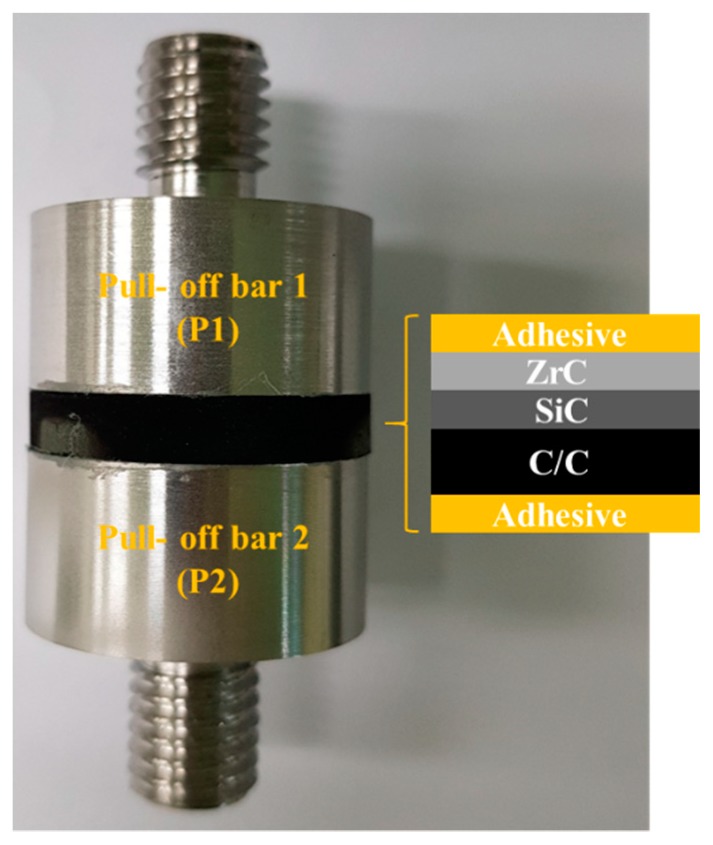
A schematic of the adhesion test.

**Figure 4 materials-12-00747-f004:**
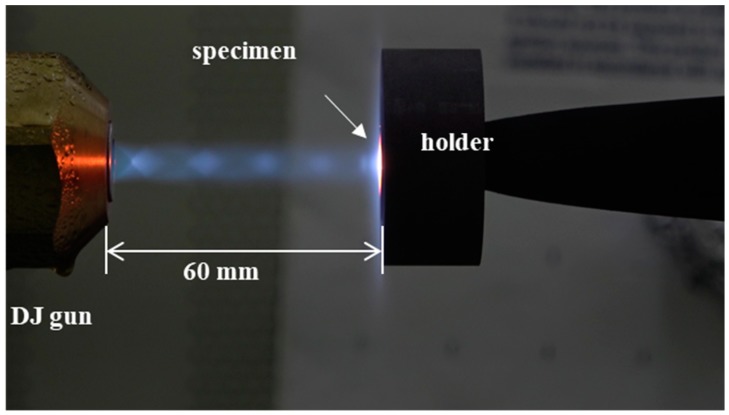
Test specimen mounted on holder.

**Figure 5 materials-12-00747-f005:**
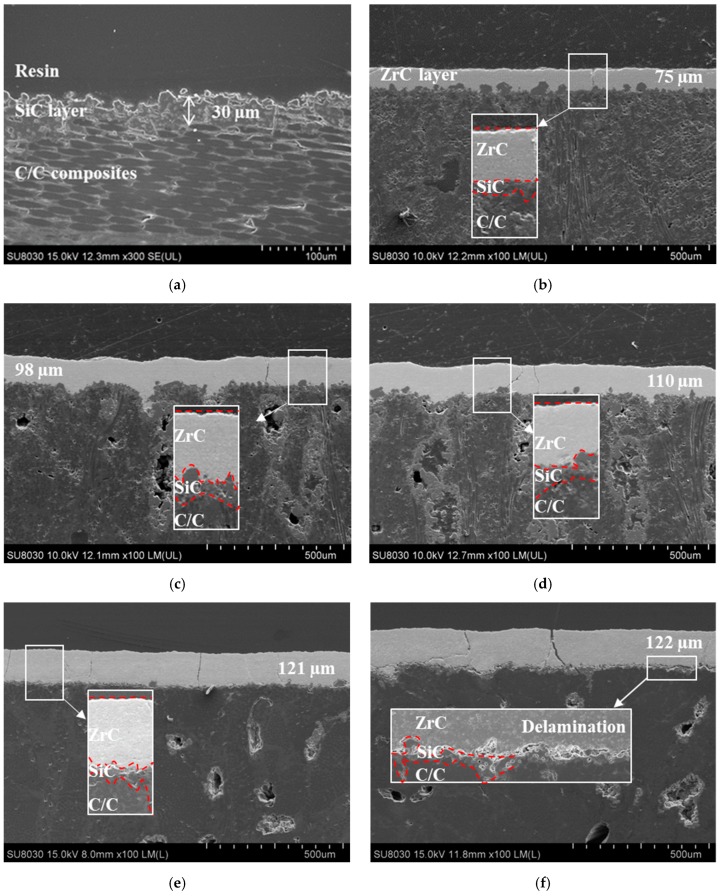
Cross-section morphology of the substrate and as-sprayed coating: (**a**) Substrate, (**b**) ZS-1, (**c**) ZS-2, (**d**) ZS-3, (**e**) ZS-4, and (**f**) ZS-5.

**Figure 6 materials-12-00747-f006:**
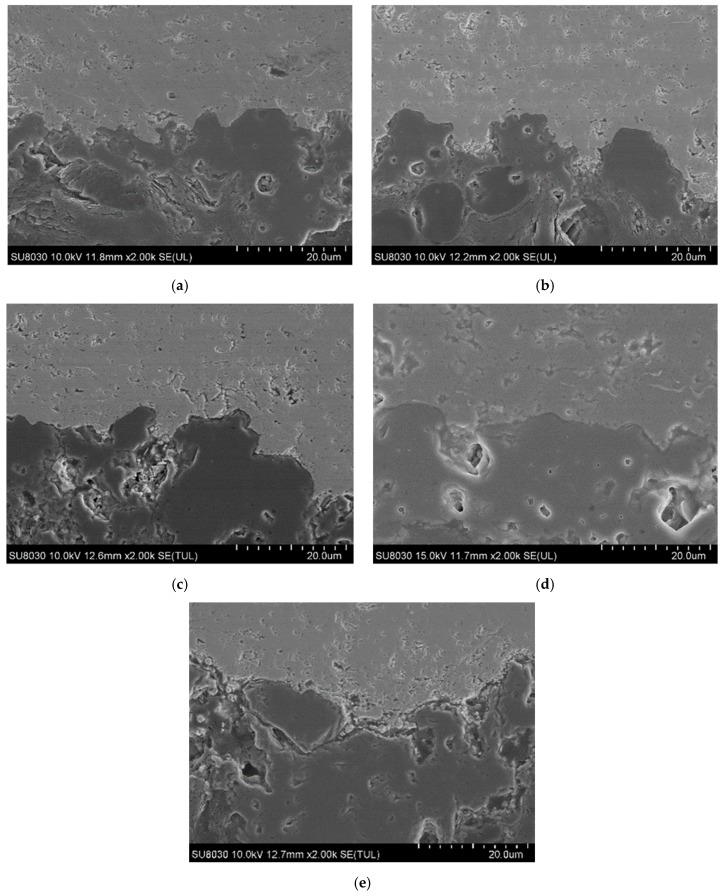
FE-SEM morphologies of the ZrC/SiC interfaces: (**a**) ZS-1, (**b**) ZS-2, (**c**) ZS-3, (**d**) ZS-4, and (**e**) ZS-5.

**Figure 7 materials-12-00747-f007:**
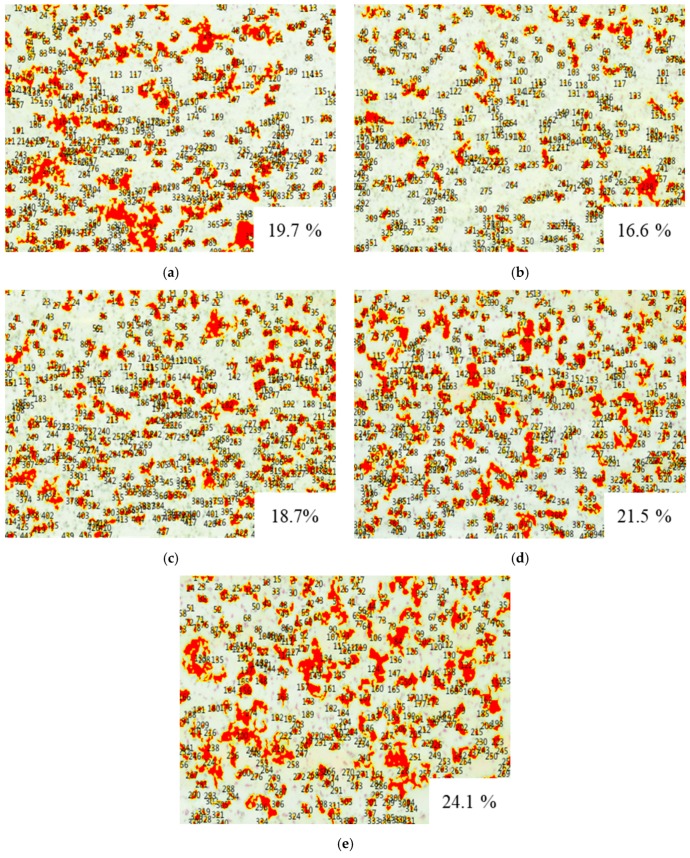
Porosity of coating layer: (**a**) ZS-1, (**b**) ZS-2, (**c**) ZS-3, (**d**) ZS-4, and (**e**) ZS-5.

**Figure 8 materials-12-00747-f008:**
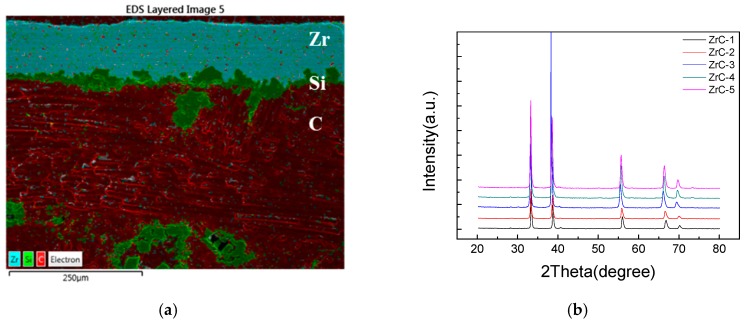
Phase analysis result: (**a**) Energy dispersive X-ray spectroscopy (EDS) mapping analysis data of ZS-2 and (**b**) XRD analysis data of as-sprayed coating.

**Figure 9 materials-12-00747-f009:**
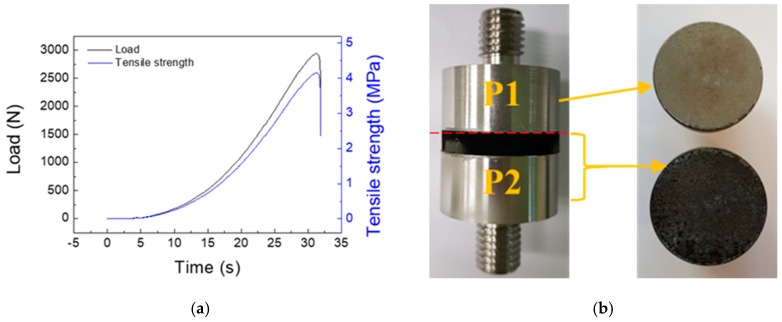
Adhesion test result of ZS-2: (**a**) Tensile strength and (**b**) morphologies after the adhesion test on the ZS-2 sample.

**Figure 10 materials-12-00747-f010:**
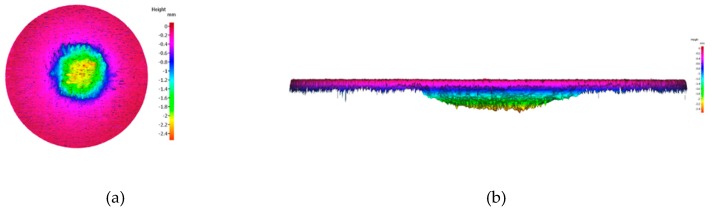
Ablation surface images for (**a**) top and (**b**) side.

**Figure 11 materials-12-00747-f011:**
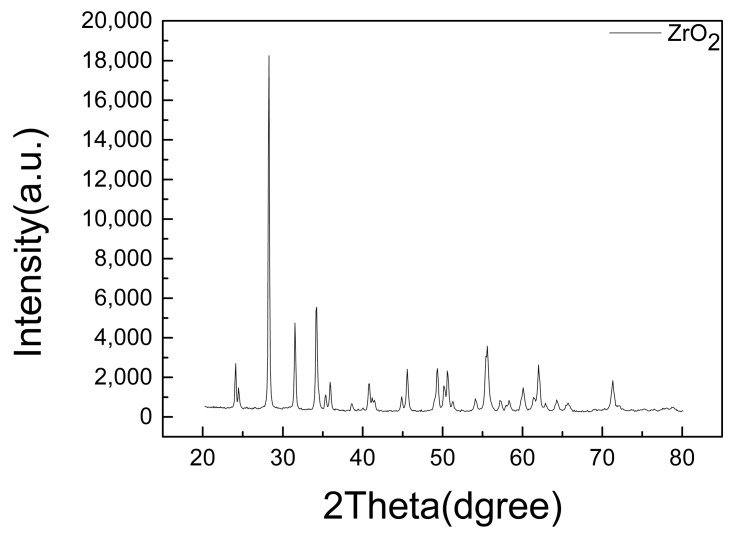
XRD analysis of ZrC coating for C/C composite after ablation for 30 s.

**Figure 12 materials-12-00747-f012:**
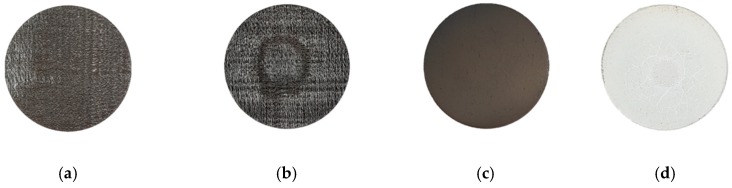
Morphologies before (**a**), (**c**) and after (**b**), (**d**) the ablation test; (**a**), (**b**) uncoated and (**c**), (**d**) coated specimens.

**Figure 13 materials-12-00747-f013:**
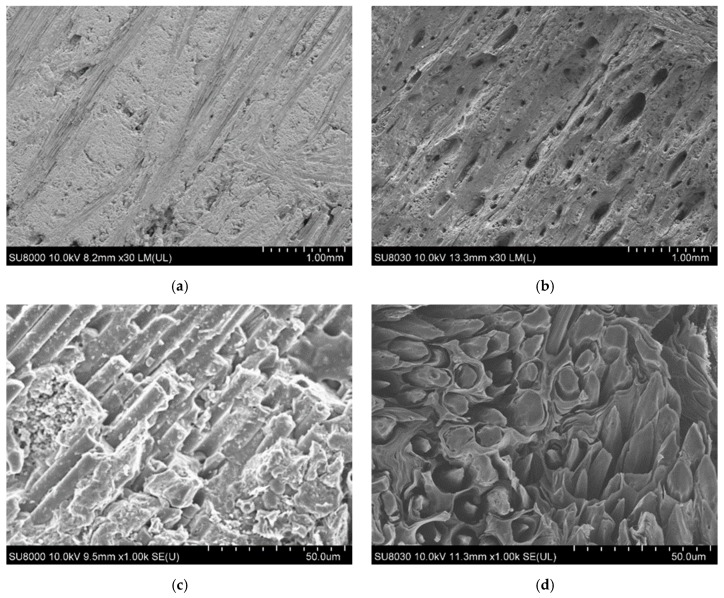
Before (**a**) and(**c**), after (**b**) and (**d**), ablation surface morphology of the C/C composite.

**Figure 14 materials-12-00747-f014:**
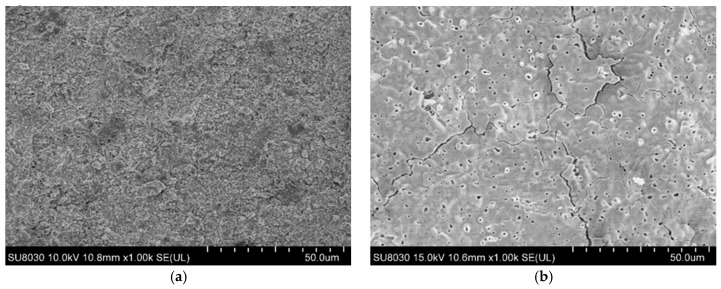
(**a**) Surface morphologies of the ZrC-coated sample before, (**b**) and after the ablation test.

**Figure 15 materials-12-00747-f015:**
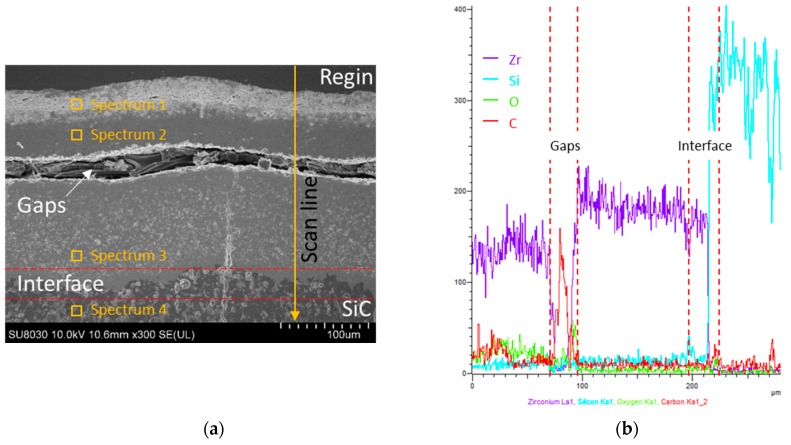
Cross-sectional FE-SEM images and EDS analysis of the ZrC-coated samples: (**a**) Cross-section image, (**b**) Line scan profile, and (**c**) Point analysis.

**Table 1 materials-12-00747-t001:** Details of the vacuum plasma spraying (VPS) parameters for ZrC coating.

Parameters		
**Pre/post-heating**	Spraying current, A	500
	Ar gas flow, NLPM ^1^	30
	H_2_ gas flow, NLPM	2
	Chamber Pressure, mbar	100
	Spraying distance, mm	210
**Coating**	Spraying current, A	500–700
	Ar gas flow, NLPM	50
	H_2_ gas flow, NLPM	10
	Feeding rate, g/min	4.5
	Chamber Pressure, mbar	50
	Spraying distance, mm	350

^1^ Normal liter per minute.

**Table 2 materials-12-00747-t002:** Ablation properties of the carbon/carbon (C/C) composite with or without ZrC coating.

Samples	Mass ablation rate $\left(\times 10^{-2}\;g/s\right)$	Linear ablation rate $\left(\times 10^{-3}\;\mathrm{mm}/s\right)$
C/C	3.21	7.73
ZrC/SiC-coated C/C	−2.67	−3.76

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
