# Peer review of "Characteristics of ZrC Barrier Coating on SiC-Coated Carbon/Carbon Composite Developed by Thermal Spray Process"

_materials, 2019, doi:10.3390/ma12050747_

Reviewer 1 Report

This manuscript presents the vacuum plasma spraying (VPS) of ZrC as protective layer on top of SiC buffer layer on carbon/carbon composite. This reviewer cannot recommend the current manuscript for publication in Materials because of the following reasons:

The title and abstract does not mention the existence of SiC buffer layer, which is a important fact. Please correct this.  

On line 54: the literature review misses many key works in thermal plasma spraying of ZrC/SiC/carbon-carbon composites. A  quick search on Google Scholar reveals: https://scholar.google.com/scholar?hl=en&as_sdt=0%2C5&q=zrc+thermal+spraying+sic&btnG=

On line 64: What is the novelty of this work? The authors need to conduct proper literature review as suggested in comment #2 to come up with the best answer.

On line 90: What is the advantage of making S-shaped path?

In Table 1, please define NLPM.

On line 138, please change the temperature to Celcius to be consistent.

In Figure 4(b)-(f), where is the SiC layer?

Based on images from Figure 4 and 5, this reviewer is concerned with the quality of ZrC films being deposited on top of SiC films. It seems that the adhesion between ZrC and SiC is very weak. This is evidence from Figure 12 and 13(a) after the ablation test. In other words, readers may think that the current ZrC film will not perform its function as protective coat effectively, as claimed in conclusion.   

Author Response

1.1 The title and abstract does not mention the existence of SiC buffer layer, which is a important fact. Please correct this.

Response: We agree with your opinion. We changed the title and mentioned the SiC buffer layer in the abstract.

1.2 On line 54: the literature review misses many key works in thermal plasma spraying of ZrC/SiC/carbon-carbon composites. A quick search on Google Scholar reveals: https://scholar.google.com/scholar?hl=en&as_sdt=0%2C5&q=zrc+thermal+spraying+sic&btnG=

Response: About line 54, Plasma spraying is one of the promising coating methods because it is economical and can be easily applied on the industrial scale -> This is generally related to commercial plasma spraying and we added references of plasma spraying applications. In addition, following your opinion, we added the references related to "Thermal plasma spraying of ZrC / SiC / carbon-carbon composites" and the manuscript was revised.

-Before: “Plasma spraying is one of the promising coating methods because it is economical and can be easily applied on the industrial scale. Vacuum plasma spray (VPS) coatings are especially preferred because of the high-purity coatings with low porosity and high deposition rate.”           

-After: “Plasma spraying is one of the promising coating methods because it is economical and can be easily applied on the industrial scale [15-17]. It is also the suitable technique for depositing carbide with high melting point such as ZrC, HfC, etc [18-19].”

1.3 On line 64: What is the novelty of this work? The authors need to conduct proper literature review as suggested in comment #2 to come up with the best answer.

Response: We already explained the reason in “introduction” part as below, we added and revised some sentences for clear understanding. 

-Before: “In VPS, the coating layer is formed by melting the constituent particles at a very high temperature and by colliding them with the base material; therefore, the inherent properties of both components, i.e., the molten particles and base substrate, are preserved. However, as numerous parameters contribute to the formation of a high-quality coating layer, it is very important to determine the optimal process parameters. Among the operational parameters, it is well known that the velocity and temperature of the plasma jet, which can be mainly controlled by the discharge current, are closely associated with the phase change, thickness and porosity.”

-After: “In APS process or other plasma spray process in atmosphere, ZrC powder is oxidized to result in the formation of zirconium oxides [20-23]. Therefore, vacuum plasma spray (VPS) coatings are especially preferred because of the high-purity coatings with low porosity and high deposition rate without forming oxides [24]. In VPS, the coating layer is formed by melting the constituent particles at a very high temperature and by colliding them with the base material; therefore, the inherent properties of both components, i.e., the molten particles and base substrate, are preserved. However, as numerous parameters contribute to the formation of a high-quality coating layer, it is very important to determine the optimal process parameters. Among the operational parameters, it is well known that the velocity and temperature of the plasma jet, which can be mainly controlled by the discharge current, are closely associated with the phase change, thickness and porosity. As mentioned above, although the coating process using VPS has many advantages, few research articles have reported on UHTCs coatings prepared using VPS technique.”

1.4 On line 90: What is the advantage of making S-shaped path?

Response: The first is to reduce the load of the robot performing the coating, and the second is to coat the surface of the sample with a uniform thickness. If the coating is performed as shown in Fig. (B), only the center of the sample can be thickly coated. Therefore, S-shaped path were used and the path shown in (a).

1.5 In Table 1, please define NLPM.

Response: We define NLPM and marked footnote at the bottom of the table 1.

-Before:

Parameters

Pre/post-heating

Spraying current, A

500

Ar gas flow, NLPM

30

H2 gas flow, NLPM

2

Chamber Pressure, mbar

100

Spraying distance, mm

210

Coating

Spraying current, A

500 - 700

Ar gas flow, NLPM

50

H2 gas flow, NLPM

10

Feeding rate, g/min

4.5

Chamber Pressure, mbar

50

Spraying distance, mm

350

-After:

Parameters

Pre/post-heating

Spraying current, A

500

Ar gas flow, NLPM 1

30

H2 gas flow, NLPM

2

Chamber Pressure, mbar

100

Spraying distance, mm

210

Coating

Spraying current, A

500 - 700

Ar gas flow, NLPM

50

H2 gas flow, NLPM

10

Feeding rate, g/min

4.5

Chamber Pressure, mbar

50

Spraying distance, mm

350

1 Normal liter per minute.

1.6 On line 138, please change the temperature to Celcius to be consistent

Response: We changed to use the same temperature unit. 

-Before: The temperature of the generated flame approaches 3,000 K, and the flame is accelerated through convergent/divergent nozzles to create a supersonic flame. 

-After: The temperature of the generated flame approaches 2730 °C, and the flame is accelerated through convergent/divergent nozzles to create a supersonic flame.

1.7 In Figure 4(b)-(f), where is the SiC layer? 

Response: We revised the images you mentioned as below to help other reviewers understand easily. As shown in figure 4(b)-(f), we marked the SiC layer in FE-SEM images as text. 

1.8 Based on images from Figure 4 and 5, this reviewer is concerned with the quality of ZrC films being deposited on top of SiC films. It seems that the adhesion between ZrC and SiC is very weak (1). This is evidence from Figure 12 and 13(a) after the ablation test. In other words, readers may think that the current ZrC film will not perform its function as protective coat effectively, as claimed in conclusion (2).  

Response:

(1) We agree with that. But, for the part you are concerned with “the quality of ZrC films being deposited on top of SiC films”, we already referred the “Results and discussion” part as next. “ZrC is uniformly coated on top of SiC without any loss of SiC, but some cracks formed~”, “~a partial delamination from the substrate is observed in a thicker case as seen in Figure 4(f)”. Moreover, it is necessary not for the ZrC layer to be well bonded to the SiC layer in all 5 samples.  This is because only the best one of the five samples is selected for ablation testing. Except for figure 4(f), it seems to be well bonded to the image as well. And we regarded the ZC-2 samples as the most appropriate for ablation test.

About the adehesion between ZrC and SiC, we added more information to help understanding our opinion. We tried to find the method of adhesion test in thermal sprayed specimens. Our opinion was supported by a graph of the tensile stress test as follows. But it is not included in the manuscript due to insufficient the data. We concluded that the ZrC layer and the SiC layer are well bonded except for Figure 4(f). The results were attached.

The schematic below is a preparation for the adhesion test. (The tensile strength test method)

As a result, between the ZrC layer and the SiC layer were not separated and between the p1 and the ZrC layer were separated. (The ZrC layer was not peeled off.) Based on these results, we concluded that ZrC is well bonded with SiC.

(2) In figure 12(b), many pores and cracks are observed from the surface after ablation, but it is a common phenomenon after the ablation test. These can be found that we already referred the reference (No. 22, 23) in the manuscript. 

< References in manuscript>

[22]. Wen, B.; Ma, Z.; Liu, Y.; Wang, F.; Cai, H.; Gao, L. Supersonic flame ablation resistance of W/ZrC coating deposited on C/SiC composites by atmosphere plasma spraying. Ceram. Int, 2014, 40, 11825-11830.

: “After the 150 mm ablation (Fig. 5b), some micro-cracks and micro-holes were found in the surface. These micro-cracks resulted from the fast cooling after ablation.”

[23]. Jia, Y.; Li, H.; Feng, L.; Sun, J.; Li, K.; Fu, Q. Ablation behavior of rare earth La-modified ZrC coating for SiC-coated carbon/carbon composites under an oxyacetylene torch. Corros. Sci, 2016, 104, 61-70.

: “Moreover, ZrO2 will undergo several phase transformations upon cooling from the melt [18], resulting in the volume expansion and the crack formation after oxidation or ablation.”

Therefore, we did not think that the adhesion of ZrC and SiC were affected the result in figure 12(b). And some references {No. 23(figure 10), 25(figure 11)} also shown the relative gaps we observed in figure 13(a). Figure 13(a) shows the cross section of the coated specimen after the ablation test.

< References in manuscript>

25. Jia, Y.; Li, H.; Fu, Q.; Zhao, Z.; Sun, J. Ablation resistance of supersonic-atmosphere-plasma-spraying ZrC coating doped with ZrO2 for SiC-coated carbon/carbon composites. Corros. Sci, 2007, 123, 40-54.

However, the gaps in two references mentioned above was observed between SiC layer and the coating layer, but in figure 13(a) of our manuscript, between SiC layer and the coating layer was not observed the gaps. Just near the oxidized top of ZrC layer was peeled off because the ZrC layer is weakened due to the oxidation of ZrC. Moreover, the interface between ZrC and SiC layer no delamination was observed. We added some sentences for clear understanding.

-Before: Part of the coating layer appears to be exfoliated when the specimen was cut for observation.

-After: Part of the coating layer appears to be exfoliated when the specimen was cut for observation. However, the bottom of Figure 13(a) shows that remained ZrC layer is still well bonded on the SiC layer.

And even after the ablation test, the unoxidized ZrC layer remained clearly. This is evidence that ZrC coated by VPS is well protected the C/C from ablative environment.

Reviewer 2 Report

Dear Authors, 

I have read your manuscript carefully and I would say that this manuscript would be very interesting for readers. Thermal barrier materials/coatings are very important from the development of the materials science point of view and constitute an important research area and what is more, now many projects are being evaluated. The objectives of the study are clearly defined. The introduction provides a good, generalized background of the topic. The results are clearly explained and are presented in an appropriate format. The figures show essential data; some of the data are also summarized in the text. I do not think any additional graphics are necessary. The cited literature is relevant to the study and balanced. This manuscript was very well prepared and could be published after correcting some linguistic and editing errors as well as adding some essential missing information. 

When the authors indicate the materials and/or devices, they should use a consistent scheme. It means, that means if the authors present the country of origin of some materials and/or devices, they should add the country of origin in cases where this information was not provided. Another option is to delete the country of origin of materials and/or devices in the whole text of the manuscript. 

The authors should use the same unit of temperature, °C or K. Now, both units are used.

Tags of the detected phases should be included in the X-ray spectra presented in Figure 7(b), as in Figure 9.

“A?” is written In Table 2, please correct it.

Author Response

2.1 The authors should use the same unit of temperature, °C or K. Now, both units are used.

Response: According to your comment, we changed to use the same temperature unit.

2.2 Tags of the detected phases should be included in the X-ray spectra presented in Figure 7(b), as in Figure 9.

Response: We revised the x-ray spectra graph.

2.3 “A?” is written In Table 2, please correct it.

Response: We corrected the wrong words.

-Before:

Samples

Mass ablation rate

(  g/s)

Linear ablation rate

( mm/s)

C/C

3.21

7.73

ZrC/SiC coated C/C

-2.67

-3.76

-After:

Samples

Mass ablation rate

( g/s)

Linear ablation rate

( mm/s)

C/C

3.21

7.73

ZrC/SiC coated C/C

-2.67

-3.76

Reviewer 3 Report

This study is very well documented, presented and clearly explained.

I only have 2 questions/recommendations:

-        The choice of ZrC is not enough argued. Is it only due to high melting point or is it another reason? Indeed, many ceramics present high melting point, higher than 3000°C. Is it due to the SiC layer?

-        Is there absolutely no interest in determining the mechanical properties of this coating? The SiC and ZrC layers could highly increase the hardness and wear resistance. Moreover, this properties could lead to open the field of possible applications?

Author Response

3.1 The choice of ZrC is not enough argued. Is it only due to high melting point or is it another reason? Indeed, many ceramics present high melting point, higher than 3000°C. Is it due to the SiC layer?

Response: At the start of this study, we was concerned about selecting some refractory carbides. Carbides have higher melting point than other refractory materials [1-2]; and they are reported to have lower vapor pressures [3]. So far, research on HfC has been done most and many papers on HfC coating method using VPS have been published. In the case of TaC, a preliminary test was performed, but there was a big problem in that TaC was decomposition into Ta2C. And we did not find a proper way to prevent it. Of course, we thought that ZrC was the most suitable candidate even if we compared to other candidates. And we chose the ZrC because it was considered that there was not sufficient data on coating with VPS. The SiC layer is only a buffer layer that reduces the thermal expansion coefficient between ZrC and C/C and improves the adhesion. There have been many reports of coating SiC before coating UHTCs on C/C [4-5].

3.2 Is there absolutely no interest in determining the mechanical properties of this coating? The SiC and ZrC layers could highly increase the hardness and wear resistance. Moreover, this properties could lead to open the field of possible applications?

Response: This study focused on protection for C/C at extreme environment. In the future, we will conduct experiments to improve hardness and porosity, and we will carry out the test on adhesion strength mentioned above. Therefore, we will apply it to more fields through the coating technique that can control the mechanical properties.

Justin, J.; Jankowiak, A. Ultra high temperature ceramics: Densification, properties and thermal stability. AerospaceLab 2011, 3, 1-11.

Wang, S. L.; Li, K. Z.; Li, H. J.; Zhang, Y. L.; Wang, Y. J. Effects of microstructures on the ablation behaviors of ZrC deposited by CVD. Surf. Coat. Technol. 2014, 240, 450-455.

Xiong, X.; Wang, Y. L.; Li, G. D.; Chen, Z. K.; Sun, W.; Wang, Z. S. HfC/ZrC ablation protective coating for carbon/carbon composites. Corros. Sci. 2013, 77, 25-30.

Wang, Y. J.; Li, H. J.; Fu, Q. G.; Wu, H.; Yao, D. J.; Wei, B. B. Ablative property of HfC-based multilayer coating for C/C composites under oxy-acetylene torch. Appl. Surf. Sci. 2011, 257, 4760-4763.

Ren, J.; Zhang, Y., Zhang; P., Li, T.; Hu, H. UHTC coating reinforced by HfC nanowires against ablation for C/C composites. Surf. Coat. Technol. 2017, 311, 191-198.

Round  2

Reviewer 1 Report

The authors have addressed most of my concerns. These are the follow up comments:

Line 123: Please provide information for EDS equipment.

Line 189: Figure 1?

Line 263: in order to fully convince the readers, peeling or tensile test as mentioned in authors' responses for sample ZS-2 should be provided.

Author Response

Reviewer #1
2.1 Line 123: Please provide information for EDS equipment.

Response: We added the information for EDS equipment.

-Before: The cross-sectional microstructure of the coating was determined by field-emission scanning electron microscopy (FE-SEM; SU-8030, Hitachi, Japan), and the components of the coating layer were analyzed by energy-dispersive X-ray spectroscopy (EDS).

 -After: “The cross-sectional microstructure of the coating was determined by field-emission scanning electron microscopy (FE-SEM; SU-8030, Hitachi, Japan), and the components of the coating layer were analyzed by energy-dispersive X-ray spectroscopy (EDS; X-MaxN80, Horiba, Japan).”

2.2 Line 189: Figure 1?

Response: We revised wrong word as below.

-Before: “As shown in Figure 1, it is clear that there are strong peaks corresponding to ZrC, while diffraction peaks of carbon are not detected, indicating that the thick ZrC layer is formed on C/C composite.”

-After: ”As shown in Figure 7(b), it is clear that there are strong peaks corresponding to ZrC, while diffraction peaks of carbon are not detected, indicating that the thick ZrC layer is formed on C/C composite.”

2.3 Line 263: in order to fully convince the readers, peeling or tensile test as mentioned in authors' responses for sample ZS-2 should be provided.

Response: We agree with your opinion. We added some sentence and adhesion test result of sample ZS-2 was inserted in the manuscript.

-Before: “Finally, the ablation test was carried out for ZrC coated sample, which was selected based on the best result in metallurgical analysis.”

-After: “Finally, the ablation test was carried out for ZrC coated sample, which was selected based on the best result in metallurgical analysis. Before the ablation test, an adhesion test of the selected sample was also carried out by universal testing machine (UTM; 5982, Instron, USA). Figure 3 shows a schematic of the adhesion test and the sample was fixed with a glue (Fusionbond 370, Hernon, USA) between the bars.”

-Before: “To perform the ablation test, we selected the ZS-2 condition because of the low porosity and uniform thickness obtained without cracks and delamination. Using this condition, we increased the number of coating cycles to obtain thicker layers up to 163 μm, which should be useful for prolonged ablation.”

-After: “To perform the ablation test, we selected the ZS-2 condition because of the low porosity and uniform thickness obtained without cracks and delamination. Using this condition, we increased the number of coating cycles to obtain thicker layers up to 163 μm, which should be useful for prolonged ablation. Figure 8 shows the results of the adhesion test on ZS-2 sample. Figure (a) is shown tensile strength and (b) is shown tested sample ZS-2. As a result, the ZrC layer was not separated with the SiC layer, but it was separated with the P1. Further, the ZrC layer was not peeled from the substrate. Based on these results, it can be seen that ZrC is well bonded with SiC.”
